# Provision of Microbiology, Infection Services and Antimicrobial Stewardship in Intensive Care: A Survey across the Critical Care Networks in England and Wales

**DOI:** 10.3390/antibiotics12040768

**Published:** 2023-04-17

**Authors:** Tim Catton, Helen Umpleby, Ahilanandan Dushianthan, Kordo Saeed

**Affiliations:** 1General Intensive Care Unit, University Hospital Southampton NHS Foundation Trust, Tremona Road, Southampton SO16 6YD, UK; 2Department of Infection, Hampshire Hospitals NHS Foundation Trust, Royal Hampshire County Hospital, Romsey Road, Winchester SO22 5DG, UK; 3NIHR Southampton Clinical Research Facility and NIHR Southampton Biomedical Research Centre, University Hospital Southampton NHS Foundation Trust, and the University of Southampton, Tremona Road, Southampton SO16 6YD, UK; 4Faculty of Medicine, University of Southampton, Tremona Road, Southampton SO16 6YD, UK; 5Department of Infection, University Hospital Southampton NHS Foundation Trust, Tremona Road, Southampton SO16 6YD, UK

**Keywords:** intensive care unit, antibiotic, infection, antibiotic stewardship, diagnostics

## Abstract

Infection rounds in Intensive Care Units (ICU) can impact antimicrobial stewardship (AMS). The aim of this survey was to assess the availability of microbiology, infection, AMS services, and antimicrobial prescribing practices in the UK ICUs. An online questionnaire was sent to clinical leads for ICUs in each region listed in the Critical Care Network for the UK. Out of 217 ICUs, 87 deduplicated responses from England and Wales were analyzed. Three-quarters of those who responded had a dedicated microbiologist, and 50% had a dedicated infection control prevention nurse. Infection rounds varied in their frequency, with 10% providing phone advice only. Antibiotic guidance was available in 99% of the units; only 8% of those were ICU-specific. There were variations in the availability of biomarkers & the duration of antibiotics prescribed for pneumonia (community, hospital, or ventilator), urinary, intra-abdominal, and line infections/sepsis. Antibiotic consumption data were not routinely discussed in a multi-disciplinary meeting. The electronic prescription was available in ~60% and local antibiotic surveillance data in only 47% of ICUs. The survey highlights variations in practice and AMS services and may offer the opportunity to further collaborations and share learnings to support the safe use of antimicrobials in the ICU.

## 1. Introduction

Antimicrobial therapy is a cornerstone in treating critically ill patients presenting with suspected infection. The surviving sepsis guidelines highlight the importance of prompt, appropriate antibiotics in such patients [1]. Antibiotic use is high in Intensive Care Units (ICU); an international point prevalence study identified over 70% of ICU patients were prescribed antibiotics [2]. Whilst antimicrobial therapy is a critical component of ICU treatment, it is important to recognize a range of potential deleterious consequences associated with antibiotic use, including antimicrobial resistance (AMR) [3].

International healthcare organizations, including the World Health Organization (WHO) recommend that antimicrobial stewardship (AMS) activities are crucial to tackling AMR [4]. Multidisciplinary teamwork in ICU, including infection specialists, has direct, measurable benefits with regard to fewer days of antibiotic therapy and attendant reduced drug expense. There are also theoretical benefits which include reduced risk of selection of resistant bacteria, fewer drug-related adverse effects, assistance in clarification of the clinical illness, and the opportunity to teach in the course of patient care [5]. It remains unclear to what extent AMS programs are adopted within the ICU and the level of variation in antibiotic prescription practices in the UK. The aim of this survey is to assess the following practices in ICUs across the UK:Microbiology services and surveillance: access to and input by microbiology and infection prevention and control specialists.Antibiotic prescription practices.The availability and use of diagnostics and biomarkers to influence antibiotic de-escalation and duration of antibiotics for common infections leading to ICU admissions.Availability of data management systems related to antibiotic use.

## 2. Methods

The survey questionnaire was developed by members of the intensive care and microbiology team using an online survey engine [6]. This survey was conducted as part of a quality improvement project by the University Hospital Southampton NHS Foundation Trust. The survey consisted of 16 questions relating to the type of ICU and microbiology/antibiotic practices. The research group was comprised of a mix of intensive care and microbiology specialists who designed the survey to assess the three core components of AMS within intensive care practices. Firstly, the availability of microbiology services and surveillance data to ICU clinicians. Secondly, questions relating to general prescribing practices within the ICU, including empirical antibiotic prescribing and the use of electronic prescribing. Questions were targeted at the duration of antibiotic therapy for certain conditions in the ICU and the use of biomarkers.

The survey was reviewed by several intensive care consultants to confirm that the questions were both clear and appropriate before being disseminated. The hospital ethics department was approached and deemed ethics approval not to be required.

The UK Critical Care Networks for England, Wales, and Northern Ireland (Adults) [7] were approached directly and through regional administrators for their members’ participation. Administrators for each of the 21 regions within England, Wales, and Northern Island were contacted and asked to forward the survey to intensive care consultants within their region. We also approached individual hospitals listed within each network for their participation with communication with individual departmental secretaries to forward the survey to the clinical lead for ICU. A link was generated for the survey, which was distributed via email. The survey was live for a period of 6 months, from February 2022 to July 2022.

At the end of the 6-month period, the survey responses were assessed. In centers where multiple responses were received, the first response was retained, and further responses were not included. Following this process, analysis of the responses was completed.

The themes covered by the survey can be viewed in (Appendix A). Most questions had a dropdown menu or multiple-choice answers with an option for freestyle comments. The summarized data are presented as numerical values and percentages.

## 3. Results

### 3.1. Demographics

A total of 101 responses were received from the 217 individual ICUs across England and Wales. Of the 101 responses, there were 14 duplicate responses from a second or third clinician responding from the same ICU. This led to a total of 87 responses from the 217 ICUs listed in the 2021 Directory of Critical Care Networks for adults giving an overall response rate of 40%.

Of the 21 areas across the critical care network, there was significant variation in the response rate. 93% of responses were from regions within England, and the remaining 7% were from Wales. There were no responses from Northern Ireland and Scotland. 91% of respondents were consultants, 8% were pharmacists, and the remaining 1% were non-consultant level doctors. 80% of responses were from clinicians working in general ICUs, with 20% from specialist ICUs (Neuroscience 5%, Cardiac 2%, others including liver, surgical, and orthopedic 13%). The participated ICUs varied in size, with a bed capacity of fewer than 15 beds (49%), 16–30 beds (30%), 31–45 beds (18%), and more than 45 beds (2%).

### 3.2. Availability of Microbiology Services and Surveillance

The survey showed that 87.4% of ICUs used hospital or trust-wide antimicrobial guidance to aid the empirical choice of antibiotic. Only 8% had local ICU-specific antibiotic guidance (Figure 1A). Almost half (46.5%) of respondents stated that they have access to local bacteria epidemiology data, including susceptibility profile and multi-drug resistant bacterial prevalence rates (E.g., methicillin-resistant Staphylococcus aureus (MRSA), extended-spectrum beta-lactamase (ESBL), carbapenemase-producing enterobacterales (CPE). Around 36% said they had no access to these data (Figure 1B).

Most units (76%) had a dedicated microbiologist or infectious diseases specialist available. This input was provided daily by 48.3%, three times weekly by 20.7%, and 10.3% had no dedicated specialist input but had specialist infection telephone advice when needed. Just over 46% of respondents had a dedicated infection prevention specialist nurse available in their unit (Figure 1C,D).

### 3.3. Antibiotic Prescribing

The majority (59%) of respondents reported having electronic antibiotic prescribing, 37% were non-electronic prescribing, 1% were unsure, and 3% were in the process of acquiring electronic antibiotic prescribing platforms. Whilst 64% prescribe empirical antibiotics with a limit on duration, 20% reported that some, but not all, empirical antibiotics were duration limited, with 16% saying there was no duration restriction when prescribing empirical antibiotics. Free text comments included: antibiotic duration was usually discussed with the microbiologist, empirical antibiotic prescriptions usually had a review date but not a stop date, while others had formal stop dates for all antibiotics.

We asked regarding the standard duration of antibiotics for common ICU infections, including septic shock, community-acquired pneumonia, ventilator-associated pneumonia, hospital-acquired pneumonia, proven line-related infection, primary intraabdominal sepsis, and community-acquired upper and lower urinary tract infections. For these eight infections surveyed, there was a range of antibiotic duration from 1–3 days to 11–14 days. The common duration of antibiotics was 7 days. Details of the standard duration of antibiotics prescribed are presented in Table 1.

### 3.4. Local Antibiotic Consumption Data

We asked about the accessibility to local antibiotic consumption data and whether this data were routinely discussed in a multi-disciplinary meeting (MDT). Data were available and discussed in a multi-disciplinary meeting in 13% of ICUs surveyed, 23% had data available, but this was not discussed in an MDT setting; 24% of data were not available, and 33% were unsure if they had access to local antibiotic consumption data. 7% selected others, and comments included that only pharmacists and microbiology staff had access to this data.

### 3.5. Routine Access to Biomarkers and Rapid Molecular Diagnostics

All respondents had routine access to biomarkers that may guide antibiotic prescription and duration of therapy Figure 2.

### 3.6. Audit and Participation in Clinical Research

We surveyed if data management systems were available to audit antibiotic use either from electronic or paper-based clinical notes. 36% reported that an electronic case-based data collection is available, 25% have manual case-based data collection, 21% have no data collection, and 22% were unsure. When asked about participation in clinical research, 79% said they participated in ICU clinical research, 14% were not participating in any research, and the remainder were unsure.

## 4. Discussion

Early and appropriate treatment of infections in critically unwell patients is crucial to reduce morbidity and mortality. However, overuse of antimicrobials can be associated with increasing resistance (AMR), a rapidly growing global problem. The World Health Organization has advised setting up antimicrobial stewardship programs to help combat AMR [8]. It encompasses a range of measures, including the appropriate choice of empiric antimicrobials aided by local guidelines and epidemiological data, timely de-escalation with the use of biomarkers, regular review by infection specialists and microbiology results, and input from infection prevention and control specialists to prevent the spread of multidrug-resistant organisms. As far as we are aware, this is the first survey of intensive care physicians assessing the availability of such services nationally, and the findings suggest a substantial variability.

Among those surveyed, the majority (87.4%) followed hospital or trust-wide antibiotic prescription guidance, and only 8% had local ICU-specific guidance. Despite being critically ill, a standard antimicrobial regimen similar to a ward-level hospitalized patient can be instituted unless there are specific risk factors or prior evidence of colonization of resistant organisms. Several observational studies have shown reduced mortality when following guidelines for community-acquired pneumonia [9,10]. For example, a patient with severe pneumococcal pneumonia and no relevant travel history can be treated based on the organism and antibiogram (or local epidemiology if there is a lack of culture sensitivities). This would typically be benzylpenicillin or amoxicillin in the UK [11]. However, local/specific ICU antibiotic guidance may be required for patients with the prolonged hospital or ICU stay with a risk of nosocomial infections from multi-drug resistant organisms. Alterations in the microbiome over time is a common issue in critical care setting due to a number of factors, including the recurrent use of broad-spectrum antibiotics [12].

In order to make an informed decision regarding the choice of antibiotic, the availability of microbiology epidemiological and/or local resistant data is essential. In this survey, 36% reported no access to epidemiological data, and 17% were unsure. In contrast, international studies report much higher accessibility to local epidemiology data (>80%) [13,14,15]. A periodical review (quarterly) of these data by a dedicated microbiology surveillance team could help assist and complement ICU antibiotic stewardship. Given that three-quarters of ICUs had a dedicated microbiologist or infectious diseases specialist input, this could be easily achievable. The National Institute for Health and Care Excellence (NICE) antimicrobial stewardship guidance advises implementing “local antimicrobial guidelines in line with national guidance and informed by local prescribing data and resistance patterns” [16].

Under half (46%) of units had access to a dedicated infection control nurse, a specialist nurse who could act as a link between ICU and the infection control team. Their role is to increase awareness of emerging infection control issues and motivate staff to improve practice [17,18]. They can also help to institute regular audit cycles of infection prevention measures. However, there is a lack of robust evidence on the effectiveness of link nurses’ programs [19]. Studies have demonstrated that simple but effective infection control strategies can reduce the transmission of MDR organisms in the ICU [20,21].

Input from a clinical microbiologist or an infectious diseases specialist can be useful in the decision-making process regarding antibiotic stewardship during the daily multidisciplinary (MDT) ward rounds. Not only facilitating stopping and de-escalation of antimicrobials but also selecting the most appropriate antimicrobial based on specific host factors. A study found input from an infection specialist was associated with significant antimicrobial modifications or discontinuation of antimicrobials. [22]. In our study, 48.3% of those surveyed had a daily ward round, and ~21% had three times weekly ward rounds. This is much higher than recently published French data, where only 9.2% and 4.6% had daily and three times weekly ward rounds, respectively [13]. Around 10% did not have a regular ward round but did have telephone advice available. It is not clear from our survey if this is the usual practice or because of recent changes due to the COVID-19 pandemic. Interaction between stewardship teams (Infection/Microbiology, pharmacists) and intensivists is key for the practical implementation of good antibiotic prescribing [23,24]. The nature and complexity of ICU patients may necessitate regular and frequent input from infection specialists to provide expert advice to a complex group with infection. Regular multidisciplinary ward rounds may also provide an educational opportunity to intensive care specialists, trainees, and nurse practitioners.

In our survey, 59% of respondents reported having access to electronic antibiotic prescribing, lower than the data from a French ICU study, which reported ~94% availability of electronic prescribing but only 54% using them [13,22]. Electronic prescribing can allow a prescriber to include a step-wise prescription pathway that consists of an indication of antibiotics, the correct dosage for the patient’s characteristics (weight, creatine clearance, etc.), and a review or a stoppage date that can be auditable easily. However, there are limitations to these electronic systems, e.g., the availability of up-to-date resources such as computers and software, and excessive electronic fields during prescription may increase fatigue among clinicians.

Sixty-four percent of respondents stated that all empirical antibiotic prescriptions are duration limited. A further 20% reported some but not all were duration limited, with 16% saying there was no duration limitation when prescribing empirical antibiotics. However, some centers commented that they had review dates but no automatic stop date. The duration varied (1–3 days, 4–5 days, or 6–7 days) depending on the source of infection. A documented planned review of the duration is an important part of antibiotic stewardship in the ICU, particularly for those with a shorter ICU stay. This is to avoid unnecessarily prolonged duration and minimize the risk of resistance development and adverse effects such as Clostridium Difficile [25]. Data from the French survey suggests 46% of ICUs reported a limitation in the duration of empirical antibiotic treatment, while in Germany, such measures were implemented by 84% [14]. Automatic stop orders have been shown to reduce antimicrobial use and antimicrobial-related adverse effects [26,27]. However, some studies have shown unintended discontinuations when using this approach [26,28]. Nevertheless, automatic stop orders can be a useful antimicrobial stewardship tool as documented in international guidance but need to ensure each case gets considered individually [29].

Guidance on the duration of antimicrobial therapy for sepsis is limited. The general recommendation (graded as weak and low-quality evidence) from the Surviving Sepsis Campaign suggests that 7–10 days of antibiotic coverage is likely sufficient for most serious infections associated with sepsis and septic shock. However, positive microbiology and/or lack of source control and neutropenia may need longer courses [30]. Our survey showed that the majority (60%) treat septic shock with a 7-day course of antibiotics. This compared to a recently published survey of health care professionals showing that the reported duration of intravenous antibiotic therapy for sepsis was variable between respondents, >10 days (17%), 7–10 days (40%), 5–7 days (27%), and 3–5 days (13%) [31].

Moreover, around 90% of responders reported an antibiotic regimen duration between 4–7 days for the treatment of community-acquired pneumonia (CAP), hospital-acquired pneumonia (HAP), and ventilator-associated pneumonia (VAP). This is reassuring and in line with the findings from a multi-center randomized controlled trial of VAP where a 5-day course of antibiotics is non-inferior to a longer course regarding clinical cure, readmissions, and mortality at 30 days [32]. Similarly, Chastre et al. showed that 8 days of antibiotic treatment for VAP was noninferior to 15 days for 28-day mortality and infection recurrence, apart from in non-fermenting coliforms like *Pseudomonas aeruginosa*; this trial excluded immunocompromised individuals [33]. The infectious disease society of America conducted a meta-analysis and found no differences between short-course antibiotic regimens (i.e., 7–8 days) and long-course regimens (i.e., 10–15 days) in terms of mortality, clinical cure, and recurrent pneumonia. Additionally, no differences were observed for pneumonia recurrence in organisms like *P. aeruginosa* with shorter courses [34].

Source control and/or intravenous administration of antimicrobials are the mainstay for the treatment of line-associated infections. The American guidelines recommend differing durations depending on the organism cultured and clinical context [35]. In our study, around 80% of responders treat line sepsis with an antibiotic duration of 1–7 days. This variable duration is probably related to the challenges associated with the diagnosis of line infections, the type of organisms cultured, and decisions on whether or when to remove the culprit lines. Additionally, the duration of therapy for intravenous catheter-related infections is still widely based on expert opinions and cohort studies rather than robust scientific evidence; inability to achieve control can lead to an increased duration of antibiotic therapy.

Surgical source control is an essential treatment for complicated primary intra-abdominal sepsis (peritonitis), both therapeutically and diagnostically. The optimal duration of definitive treatment has yet to be established. However, between 7 and 14 days have generally been an acceptable duration, and some experts recommend up to 3 weeks [36,37]. A prospective study has shown that in patients with adequate source control, outcomes with antibiotics for 4 days were similar to those with 8 days of antibiotics, but the patients from this study were not critically ill. Lack of source control can lead to extending the duration of antibiotic therapy [38]. With regards to urinary tract infections, two randomized trials in complicated urinary tract infection (UTI) have shown clinical non-inferiority of shorter quinolone courses (5–7 days) versus longer (10–14 days) with less associated collateral damage with shorter courses [39,40]. Despite the above data, critically ill patients are usually underrepresented in clinical trials evaluating the optimal duration of antibiotic treatment in organ-specific infections; hence more studies are required to define those and other infections that we commonly encounter in ICU.

With regards to the utilization of biomarkers and rapid access diagnostic platforms, all had access to C-reactive protein; 80% and 68% had access to procalcitonin (PCT) and Beta-D-glucan, respectively. There has been a marked increase in the use of PCT since the COVID-19 pandemic, from 47.6% to 84.4% [41,42]. A rapid access molecular PCR for single or multiple respiratory pathogens was available to 65%, where 49% reported they could access these results within 24 h and 16% within 2 h. These rapid access platforms may help guide infection prevention measures to minimize health care-associated transmissions and can assist stewardship and reduce the use of broad-spectrum antibiotics in ICU. However, more studies are required to determine cost-effectiveness of these interventions as well as the impact of other biomarkers, such as Pro-Adrenomedullin [43].

The survey also showed that antibiotic consumption data are not commonly discussed in a multidisciplinary setting. While these data are generally available to pharmacy departments, its accessibility to prescribers is crucial for benchmarking with comparator ICUs to improve prescribing practices, which are all part of antibiotic stewardship [44]. For the purpose of the audit, electronic case base data management systems were available for 36%, and 25% had manual data collection systems. Most centers (79%) participate in ICU research. Quantification of such digital data could help to assess antibiotic prescription practices at a regional or national scale.

Our study has several limitations. There is no detailed information about the capacity of ICUs enrolled in the study in terms of case mixes, severity, occupancy, and served populations, which can impact clinical practice. Not all centers responded to the survey request, which is a major limitation as the data we have is relatively limited. While we approached 217 centers from all current Critical Care Research Networks, only 87 (40%) provided a response. The electronic approach and approaching clinical leads for ICU rather than all intensive care doctors in each hospital may have introduced selection bias. Approaching ICU clinicians directly would probably have resulted in more responses and potentially more understanding and ownership of working within antimicrobial stewardship programs implementation initiatives. Moreover, we acknowledge that this is an in-house or self-developed survey without external testing, and most of the data was from England and may not be transferable to Northern Ireland or Scotland. Nevertheless, this is the first national intensive care survey on microbiology services availability and antibiotic stewardship.

## 5. Conclusions

The survey demonstrates significant variations in practice and service availability. Our findings clearly demonstrate that not all ICUs have regular infection or microbiology specialist multi-disciplinary ward rounds. This can be addressed through directorships or commissioning teams with clear specifications for ICU services. Societies and colleges can also provide guidance on the minimum required input from infection specialists and other multidisciplinary teams like pharmacists with clear descriptions of roles and responsibilities. Furthermore, systems and access to data regarding antibiotic consumption and resistant data should be timely not only to infection specialists or pharmacists but also to intensivists, allowing MDT decision-making with regard to empirical antibiotic choices and further guidance. This can be achieved via prior agreed governance meetings, e.g., quarterly, to discuss antibiotic consumption and antibiotic-resistant data, depending on local needs. Finally, participation in innovation, audit, quality improvement projects, and research is crucial to improve the quality of care. The development of a collaborative network of intensive care, clinical microbiology, and infectious diseases (NICCMID) specialists may help to coordinate these nationally to align the national and international strategies to improve ICU patient care while minimizing antibiotic usage and future resistance.

## Figures and Tables

**Figure 1 antibiotics-12-00768-f001:**
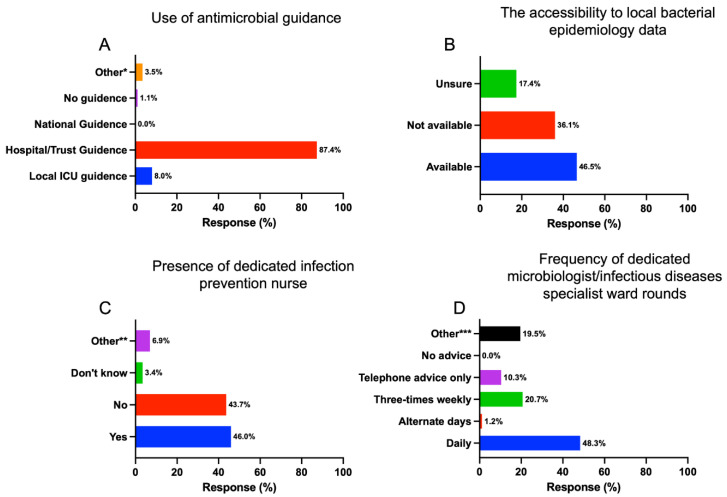
The responses for the use of antimicrobial guidance (**A**), the accessibility to local bacterial epidemiology data (**B**), the presence of a dedicated infection prevention nurse (**C**), and the frequency of specialist ward rounds (**D**). * 3.5% highlighted using all forms of guidance along with microbiology-led recommendations. ** Nearly 7% of comments highlighted the use of link nurses between ICU and infection control nurses available to the wider hospital rather than specifically dedicated to ICU; *** Circa 20% included once weekly 5 days a week and weekends by phone, telephone ward rounds mostly weekdays, a mixture of virtual and MDT ward rounds. Comments noted the use of Microsoft Teams as a platform used to complete multidisciplinary ward rounds. There were no responses highlighting no access to microbiology or infectious diseases services.

**Figure 2 antibiotics-12-00768-f002:**
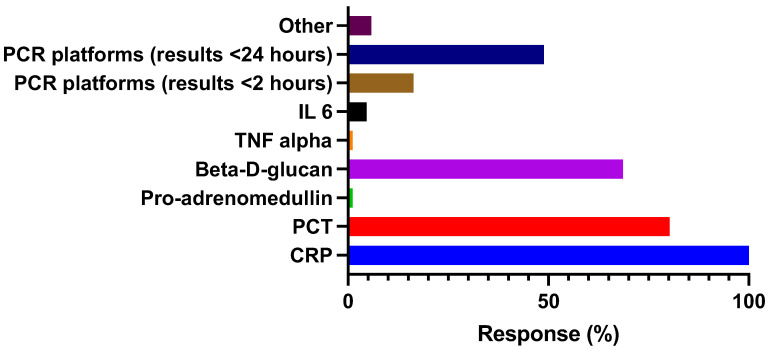
Routine access to biomarkers and rapid molecular diagnostics. CRP: C-reactive protein. PCT: Procalcitonin. IL6: Interleukin 6. TNF-alpha: (tumor necrosis factor-alpha). PCR: Rapid molecular PCR—for single or multiple respiratory pathogens.

**Table 1 antibiotics-12-00768-t001:** The standard duration of empirical antibiotics for common infections in ICU across the UK.

	1–3 Days	4–5 Days	6–7 Days	8–10 Days	11–14 Days
Septic shock	6.1%	19.5%	58.5%	11.0%	4.9%
Community acquired pneumonia	2.4%	53.0%	36.2%	6.0%	2.4%
Ventilator associated pneumonia	1.2%	50.6%	40.0%	4.7%	3.5%
Hospital acquired pneumonia	1.2%	45.9%	45.9%	4.7%	2.3%
Proven line infection *	18.8%	18.8%	42.4%	10.6%	9.4%
Primary-intraabdominal sepsis **	2.4%	33.7%	36.2%	18.1%	9.6%
Community acquired urinary sepsis (lower UTI)	22.6%	50.0%	21.4%	2.4%	3.6%
Community acquired urinary sepsis (upper UTI)	4.8%	34.5%	41.7%	8.3%	10.7%

UTI: Urinary Tract Infection. * Proven line infection—duration of antibiotics after line removal. ** Primary intra-abdominal sepsis with source control.

## Data Availability

Available on request.

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
