# Peer review of "Provision of Microbiology, Infection Services and Antimicrobial Stewardship in Intensive Care: A Survey across the Critical Care Networks in England and Wales"

_antibiotics, 2023, doi:10.3390/antibiotics12040768_

Round 1

Reviewer 1 Report

Dear authors:

I enjoyed the opportunity to review the manuscript “Provision of Microbiology, Infection Services and Antibiotic Stewardship in Intensive Care; A Survey across the Critical Care Networks in England & Wales”. This is an interesting piece of work which provides national-level data and identifies opportunities for further enhancement of antimicrobial stewardship in ICUs in England. The topic aligns with the scope of the journal. However, I have concerns about the design and quality of the survey questionnaire (see comments below) which affect the quality of data collected. Low response rate further limits the data collected. Additionally, while data presented in the manuscript were meaningful to ICUs in England, it was not clear how results would be of greater interest to other readers and how the manuscript would value-add to existing literature on antimicrobial stewardship. 

General comments

·       Be consistent in whether antimicrobial stewardship program was abbreviated as ASP or AMS. Additionally, be clear on whether this abbreviation is used in singular or plural form to ensure grammatical consistency.

Introduction

·       Page 2 lines 50-55: There might be a formatting issue. In the PDF version of the manuscript, I see phrases and sentences and it was not clear to me what they were intended to convey.

Methods

·       Please provide more details on the design and development of the survey questionnaire. How were the questions developed? How did the researchers ensure that the survey design and question phrasing were clear and appropriate?

·       Was ethics approval obtained for this study?

Discussion

·       Page 5 lines 170-172: Please provide citation(s) for this reference. I am not convinced that most ICU admission are likely to due to community-acquired infections and using a generic hospital guidance is appropriate because in my own practice and based on published literature, ICU antibiogram may demonstrate decreased susceptibilities and increased prevalence of resistant microorganisms. In many clinical practice guidelines (e.g. from IDSA), broader empiric antibiotics are recommended for critically ill patients in the ICU.

·       Page 7 lines 256-259: Given the Chastre et al. trial, do the authors imply that VAP caused by Pseudomonas aeruginosa should be treated for 15 days instead? Please update this section of the discussion in light of the updated IDSA guideline on nosocomial pneumonia from 2016.

·       Page 7 lines 260-261: Please provide citation for this sentence.

Supplementary appendix

There were typographical errors (e.g. guidence, ward around) misaligned question stem and response anchors, and ambiguous terms (e.g. what kinds of access would constitute “routine access”). Therefore, I am concerned about the design of the survey questionnaire. For example, question 5 “Do you use antibiotic guidance in the ICU”? This was phrased as a yes/no question, yet the response anchors did not match up. As another example, question 14 really included 2 questions. A better way to phrased would be to ask if respondents have access to local antibiotic consumption data (yes/no) and for those who answered yes, then ask them about whether the data would be discussed in multi-disciplinary

Reviewer 2 Report

This study is an online questionnaire targeting clinical leads for ICUs in each region listed in the Critical Care Network for the UK. This survey aimed to assess the availability of Antimicrobial stewardship services and antimicrobial prescribing practices in the UK ICUs.

Introduction

The introduction should provide information that indicates other researchers' efforts and how they were able to link the availability of these services to the effectiveness of using antibiotics in intensive care areas.

Methods

§  Line 65-68: “ Administrators for each of the 21 regions within England, Wales, and Northern Island were contacted. We also approached individual hospitals listed within each network for their participation. A link was generated for the survey which was distributed via email. The survey was live for a period of 6 months from February 2022 to July 2022.”

It is clear from the preceding text that the survey was directed at intensive care managers without addressing practitioners in these areas, which may pose a barrier to a true evaluation of real practitioners' conviction about the availability and effectiveness of the services under study.

§  The method section should include information about the questionnaire's validation and reliability testing, as well as the statistical analysis used in this study.

A brief description of the capacity of ICUs enrolled in the study is required in terms of not only beds number, but also the variety of cases, occupancy, and served populations.

Results

§  Line 80-82: “This led to a total of 87 responses from the 217 ICUs listed in the 2021 Directory of Critical Care Networks for adults giving an overall response rate of 40%.”

Do the authors believe that including practitioners in intensive care units in addition to the directors of these units would have increased the response rate, giving the study more credibility and making it more generalizable? Is there any possibility they widen the respondents' circle to include all intensivists?

General comments

§  The discussion is well-written, only there remains room to add some details regarding reviewing the opportunities and tactics of resolving the negatives that arose as a result of the study

§  Review language and grammar mistakes

Reviewer 3 Report

Whole Manuscript required a lot of technical editing. Some technical editing is needed in the Authors information section. Line 18 missing comma.

Introduction should be widened with AMS ICU practices, it is unclear why the authors just jumped to ADE.

Lines 50-55 should be presented in bullets

Development of the survey should be described in greater detail in the methods section.

Please refrain from putting links in text-put them under references and then cyte where needed

Handling duplicates should be described in the methods section

Table 1 is not needed if survey is published as supplementary data.

Line 80 needs technical editing

Line 87, do not start sentence with a number-correct in entire text

It is unfortunate that the authors did not have enough responses from all regions to do analysis according to specific region or hospital size, this would add a lot to quality and significance to the Manuscript.

I would also recommend other type of data presentation as presented figures are not exactly clear.

Overall, the manuscript just gives basic information without any comparison or new data.

Round 2

Reviewer 1 Report

Thank you for the opportunity to review revisions to the manuscript “Provision of Microbiology, Infection Services and Antibiotic Stewardship in Intensive Care; A Survey across the Critical Care Networks in England & Wales”. I appreciate the authors’ efforts in addressing my comments meticulously. My biggest concern remains the quality of the survey. As the data collection tool, quality of the data collected affects the quality of results and integrity of the study as a whole. I acknowledge that this is not something that the authors can revise at this point but this being a self-developed survey that had not undergone rigorous testing should be acknowledged as a limitation. 

Author Response

Thank you we have added the below to the limitations as suggested

“..we acknowledge that this is an in-house or self-developed survey without external testing and….”

Reviewer 2 Report

The authors responded positively to our comments... I wish this work included the greatest number of participants from intensive care physicians, as the doctors' acceptance and understanding of working within antimicrobial stewardship programs is the most significant barrier to the implementation of these initiatives.

The work is good up to this point, and there is no reason not to publish it in its current form.

Author Response

Thank you we have added the below to the limitations as this is a n important point, please see highlighted text in the manuscript 

"Approaching ICU clinicians directly would probably have resulted in more responses and potentially more understanding and ownership of working within antimicrobial stewardship programs implementation initiatives."